# Enhancing Conformational Sampling for Intrinsically Disordered and Ordered Proteins by Variational Autoencoder

**DOI:** 10.3390/ijms24086896

**Published:** 2023-04-07

**Authors:** Jun-Jie Zhu, Ning-Jie Zhang, Ting Wei, Hai-Feng Chen

**Affiliations:** 1State Key Laboratory of Microbial Metabolism, Joint International Research Laboratory of Metabolic & Developmental Sciences, Department of Bioinformatics and Biostatistics, National Experimental Teaching Center for Life Sciences and Biotechnology, School of Life Sciences and Biotechnology, Shanghai Jiao Tong University, Shanghai 200240, China; 2Shanghai Center for Bioinformation Technology, Shanghai 200240, China

**Keywords:** intrinsically disordered proteins, variational autoencoders, enhancing sampling, autoencoders

## Abstract

Intrinsically disordered proteins (IDPs) account for more than 50% of the human proteome and are closely associated with tumors, cardiovascular diseases, and neurodegeneration, which have no fixed three-dimensional structure under physiological conditions. Due to the characteristic of conformational diversity, conventional experimental methods of structural biology, such as NMR, X-ray diffraction, and CryoEM, are unable to capture conformational ensembles. Molecular dynamics (MD) simulation can sample the dynamic conformations at the atomic level, which has become an effective method for studying the structure and function of IDPs. However, the high computational cost prevents MD simulations from being widely used for IDPs conformational sampling. In recent years, significant progress has been made in artificial intelligence, which makes it possible to solve the conformational reconstruction problem of IDP with fewer computational resources. Here, based on short MD simulations of different IDPs systems, we use variational autoencoders (VAEs) to achieve the generative reconstruction of IDPs structures and include a wider range of sampled conformations from longer simulations. Compared with the generative autoencoder (AEs), VAEs add an inference layer between the encoder and decoder in the latent space, which can cover the conformational landscape of IDPs more comprehensively and achieve the effect of enhanced sampling. Through experimental verification, the Cα RMSD between VAE-generated and MD simulation sampling conformations in the 5 IDPs test systems was significantly lower than that of AE. The Spearman correlation coefficient on the structure was higher than that of AE. VAE can also achieve excellent performance regarding structured proteins. In summary, VAEs can be used to effectively sample protein structures.

## 1. Introduction

Intrinsically Disordered Proteins (IDPs) [1] have no fixed 3-dimensional structure under physiological conditions. Previous works have shown that IDP content in the eukaryotes proteomics is more than 40% [2]. Based on their variable conformations, IDPs play key roles in biological processes, such as transcription, signal transduction, and protein modification, upon binding to a wide variety of receptors or ligands [3]. Meanwhile, IDPs are closely associated with complex diseases such as tumors, cardiovascular diseases, and neurodegeneration, e.g., p53 tumor suppressor protein [4], α-synuclein [5], and abnormally phosphorylated Tau protein [6] in Alzheimer’s disease. Additionally, many disease-associated proteins often contain intrinsically disordered regions in their functional cores, which are mostly target sites of specific drug molecules.

For the study of IDPs, the critical conformations and corresponding dynamic processes are decisive factors in the results. Nuclear Magnetic Resonance (NMR), X-ray crystallography, and CryoEM are traditionally experimental methods in structural biology used to measure protein structures. However, due to the characteristic of conformational diversity, these traditional methods could not capture the conformation ensemble [7]. Molecular dynamics (MD) simulations can generate a continuous, atomic-level trajectory, which provides a powerful tool for studying the structured and dynamic properties of disordered regions [8]. Therefore, MD simulations become an important supplementary for experimental methods. Since MD simulations are of high computational cost, which restricts the simulation time scale to a few microseconds, there is still an unavoidable risk of under-sampling [9]. To obtain the IDP conformation as extensively as possible, a longer simulation as an adequate sampling strategy also means more computational resources.

Recently, deep learning has been proposed to improve the sampling efficiency of protein conformations [10]. By transforming the Cartesian coordinates of three-dimensional conformations into low-dimensional vectors, neural network models can be trained to generate the latent space from which new three-dimensional conformations can be sampled at a relatively low computational cost [11]. Gupta et al. [12] designed an autoencoder (AE) for IDP trajectories, calculating a multivariate Gaussian distribution in latent space to sample more various IDP conformations. However, according to the Spearman correlation coefficient between the AE-generated conformations and the original ones, the Gaussian distribution might not conform to the true distribution of IDP conformations. Here we designed variational autoencoders (VAEs) to avoid this contradiction and explore the conformational space of IDPs more rationally. After conducting comparison tests in all 5 IDP systems, ranging from RS1 with 24 residues to α-synuclein with 140 residues, the performance of VAEs was better than that of AEs with generated conformations more similar to the original ones.

## 2. Results

We built a VAE to generate protein conformations that are more fully reconstructed based on as few simulated conformations as possible, mainly for IDPs. The VAE model was tested on 5 IDP systems: RS1, Abeta40, PaaA2, R17, and α-synuclein. Upon training on sampled conformations from short MD simulations, VAE can effectively generate new conformations that sample from the long MD simulations.

To evaluate the performance of different IDP systems, we first calculated the RMSD between the VAE-generated protein conformations and the input ones to measure the similarity between the two. Then to represent the overall conformation distribution, the Spearman correlation was calculated. Next, we used the chemical shifts and radius of gyrations to further verify the authenticity and rationality of the structure. Moreover, we also performed the same test process on 2 structured protein systems: BPTI and ubiquitin, the results suggesting that there is still huge room to improve the current VAE model.

### 2.1. VAEs Perform Better than AEs on Different Data Split

First, to evaluate the performance of AEs and VAEs, models were trained with data of different scales in generating new IDP conformations. We split 50,000 conformations of each protein into two parts, the first part (e.g., the first 10% of the conformations) as a training set and the latter as a test set for model training. Such a splitting approach is based on the feature of MD simulation that the first part often contains more extended and diverse conformations, while the latter part contains relatively compact ones [13]. By training the model with the first part, the model could learn enough diverse conformations, and a similar training process can be accomplished on a rather short MD trajectory. For the structure of AEs, we set the latent dimension to 2/3$N_{Res}$ ($N_{Res}$: number of residues) and built them with four-layer encoders, referring to the setting designed by Gupta et al. [12]. For VAEs, as the latent dimension was proved not to significantly affect the quality of generated conformations, we set the latent dimension to 2. We built them with four-layer encoders, the same as that of AE (Table 1). From this comparative test, VAE had clear advantages against AE on the conformation generation task. The results are shown in Figure 1.

VAEs performed better than AE with lower RMSD and higher Spearman coefficients, meaning that the VAE-generated conformations could better restore the characteristics of the original ones, e.g., on the PaaA2 system, the training model with the first 50% of the data set, equal to 2.5 W conformations, VAE produced an average RMSD of 7.53 Å while AE gave 9.55 Å. The mean RMSDs and Spearman coefficients of VAE are significantly lower than those of AE in the Wilcoxon rank sum test with a *p*-value less than 1.58 × 10^−5^. Meanwhile, VAE did not sample extremely deviated conformations with large RMSD with the original set, as AE did. This is because the inference layer of VAE is based on the established distribution, which ensures more continuous sampling, e.g., for Abeta40, AE gave a maximum RMSD of 38.63 Å while the maximum RMSDs from VAE were all lower than 24 Å. In addition, VAE produced much higher Spearman coefficients than AE. In Abeta40 and PaaA2 systems, AE gave the lowest coefficients of 0.379 and 0.186, respectively, which suggests that using Gaussian distribution to sample in the latent space of AE may not accurately capture the low-dimensional distribution mapped from the encoder. On the other hand, VAE produced more uniform latent space, ensuring that sampled latent vectors could fit better the distribution of input data, the Spearman coefficient of which was always larger than 0.55.

### 2.2. VAEs Can Be Further Optimized by Adjusting Hyperparameters

After clarifying the advantages of the VAE over AE with the same layer parameters, we further adjusted the parameters of the VAE model to check whether the model could be further optimized.

Here we first adjusted the number of layers of the model. The number of neuron units in each layer was adjusted according to the number of layers to balance the compression factor between layers (Appendix A). The test results are shown in Figure 2.

For almost all neural network models, more hidden layers mean that the network structure is more complex and may obtain more comprehensive features of the input object in the training process. Still, it is also more likely to lose essential biological information due to overfitting. According to the above results, the change in the number of VAE layers did not significantly affect the RMSDs of generated IDP conformations. Only when the VAE was of 4 or 5 layers did α-synuclein perform smaller RMSDs. In view of the Spearman coefficient, 4-layer VAE reached the highest coefficient on all IDPs except for RS1. Taken all together, 4-layer VAE could perform relatively better on our IDP structure generation task.

We next examined the effect of training epochs and the dimension of latent space on VAE performance. Here we used the Spearman coefficient and the average RMSD on the test set to check the performance of VAE under each round. The training epochs and latent dimension were recorded from 2 to 20 at 2 intervals. Appendix A showed that VAEs rapidly converged within 5 epochs. Meanwhile, the dimension of latent space did not significantly affect the performance of VAE, as shown in Figure 2. VAE with 2-dimensional latent space performed slightly before others in the tests, e.g., the VAE of 2-dimensional latent space gave the minimum RMSD of 9.72 Å on Abeta40.

### 2.3. Visualization of VAE-Generated Conformations

To display representative and diverse VAE-generated conformations, we generated and refined 500 conformations for each IDP. We then clustered the conformations with k-centroids according to their RMSD. PyMol displayed the conformations identified as clustering centers (Figure 3 and Appendix A). VAE could capture global IDP characters as it generated conformations similar to input ones. VAE generated rather extended conformations for RS1 while reproducing the random coils well for α-synuclein, which means that VAE could generate disordered regions. Meanwhile, for Abeta40, PaaA2, and R17, which performed more compactly and owned several helices in a natural state, VAE-generated conformations also achieved a similar proportion of these structural regions. Although VAE mispredicted the short sheets to loops of the same direction in α-synuclein, which might result from the hardness of learning long-range interactions for deep learning models, VAE performed well on the target IDP systems overall. Nevertheless, we could find VAE-generated conformations with large RMSD against input ones far from the original set, as indicated in Figure 1. For this part, we considered that these conformations still preserved the essential structure properties of IDPs exactly, which represented novel conformations. Through conducting principal component analysis, we found that the generated conformations are reduced to quite a wide range in the first and second principal components and that they form a distribution similar to those from MD, especially in RS1 and Abeta40 in the same principal component space, which further proved generated conformations with diversity and novelty (shown in Appendix A).

### 2.4. Experimental Validation of VAE-Generated Conformations

Finally, we validated the VAE-generated conformations to check if they corresponded to relevant experimental data. We calculated secondary chemical shifts of Cα for generated configurations and the MD trajectories with SPARTA+ [14] and compared them with experimental measurements. We also calculated the radius of gyration (Rg) for each IDP on VAE-generated conformations and MD trajectories by MDTraj. We also calculated a Ramachandran plot for the generated ensembles to further check their validity and diversity (Appendix A). Figure 4 shows the comparison result for secondary chemical shifts from three datasets, and Figure 5 for Rg values. Here, 50,000 conformations are generated for chemical shift calculation, the same number of conformations with MD trajectories. The results show that the VAE-generated conformations can achieve results similar to MD simulation for local feature reduction of the experimental structure (Figure 4). The mean squared error between chemical shifts of VAE conformation and the experimental ones did not deviate from that between MD trajectories and experimental structures. Besides, Rg calculated from VAE-generated conformations performed slightly higher than that from MD trajectories in three IDP systems, RS1, Abeta40, and α-synuclein, while being consistent with that from MD trajectories in the other two systems. We can also find that in some tested protein systems, the conformations generated by MD simulation and VAE have large deviations from the experimental values, such as Rg values of PaaA2 and α-synuclein. Considering that our VAE model is trained completely based on the structures from the MD simulations, correcting the error between MD and experimental data is the fundamental strategy to further improve the performance of the VAE model.

### 2.5. Tests of VAEs on Structured Proteins

After proving that VAE performed well in enhanced conformational sampling of IDPs, we verified it on two structured proteins to check whether it could achieve the same success as IDPs. The two structured proteins are BPTI and ubiquitin, which contain 58 and 76 residues, respectively (Table 2). For structured proteins, ubiquitin is processed similarly to IDPs, simulated with force field ESFF1 and TIP3P water model. The trajectory of BPTI is taken from the previous work of D.E. Shaw et al. [15]. We conducted the same tests on structured proteins, calculating RMSD and Spearman coefficients between VAE-generated conformations and MD trajectories. Results showed that VAE performed better than AE in the conformational acquisition of structured proteins (Figure 6).

There was no significant change in their prediction effects on AE and VAE as the proportion of the training set was increased because the conformation of structured proteins was much more stable than IDPs, so the conformational distribution in the MD simulation trajectory is also more uniform. Compared with AE, VAE showed little difference in the performance of these two structured protein systems. For the BPTI system, the average RMSD of AE was about 1.42 Å, while VAE was about 1.70 Å. For the ubiquitin system, the average RMSD of AE formation and input conformation was about 1.63 Å, while that of VAE was about 1.55 Å.

From visualization and experimental validation, the difference between VAE performance for structured protein conformations is small, and the cluster for generated distribution is more concentrated. Especially for the results of the ubiquitin system, the RMSD difference between conformations was less than 0.05 Å in 92.9%.

## 3. Discussion

Extensive sampling protein conformation space is a critical foundation for molecular simulation and docking [16]. In this research, we constructed a VAE model, a generative neural network, which performed well on both IDPs and structured proteins. The VAE is trained to learn the sampled conformations from MD simulations in a relatively short period (Appendix A), which not only enables the reduction of the input protein structure with a nonlinear transformation towards low-dimensional space but also generates new conformations by disturbing the latent space with Gaussian noise [17,18]. A well-trained VAE can sample conformations of target proteins effectively and accurately, both for structured proteins and IDPs.

Compared to AE, VAE fits the latent space to a more stable distribution, like the Gaussian distribution here, which contributes to continuous sampling from the latent space to avoid generating unreasonable conformations, and also enables smoother property changes between the reconstructed structures.

In this study, the scale of VAE-generated conformations was consistent with the test set size. As VAE applies uniform sampling to the latent space, when increasing the scale of generated conformations, we can foresee that the conformational features presented in the generated structures will be more continuous [12], and some high-energy conformations different from the input ones will be predicted. It is traditionally quite hard to sample these conformations with MD simulation, though it is becoming much easier for VAE. However, it is difficult to measure the quality of this part of the generated conformation. We paid more attention to whether most generated conformations were similar to the input ones because it could indicate whether our model learned the essential features of IDP successfully. Because most conformations are similar to the input ones, the remaining conformations will likely be the potential high-energy IDP conformations obtained by changing the sampling distribution or IDP conformations in unsampled potential wells. From the results of this study, VAE can indeed restore the characteristics of input conformations in a large proportion of the generated ones (Figure 3). Therefore, the performance of VAE is better than that of AE, and the generated extreme conformations of VAE can be considered real unsampled IDP conformations.

From the tests, we found that there was no linear correlation between the VAE reconstruction mean deviation and the size of protein systems. For example, on the Abeta40 system (40 residues), the average RMSD between the reconstructed structure and the input structure reached 7.36 Å, lower than 8.66 Å from the RS1 system (24 residues). While visualizing the generated conformations, we found that structured motifs, such as α-helix or β-sheet, were preserved better than disordered regions.

Obviously, the present VAE training mode has some limitations concerning reproducing trajectory for IDPs. Though performing better than AE, VAE has only flatted atomic coordinates vectors as the input during feature extraction of the data. Some potential physical and chemical properties of proteins are also lost as the coordinates are tiled. This might be why the VAE predictive effect reaches a certain limit [19]. Therefore, to further improve the performance of the deep learning network for protein conformation generation, we can consider retaining as much protein property information as possible. We believed that the protein representation by graph network [20] is very worth trying, for which the atomic position and chemical bond information of proteins would be reduced and extracted in the form of graphs [21,22]. Therefore, the dimensionality reduction processing and reduction criteria of such graph-like data need more thought in the next step.

## 4. Material and Methods

### 4.1. Molecular Dynamic Simulation

We conducted 1 μs MD simulations for the target IDP systems, including RS1 [23], Abeta40 [24], PaaA2 [25], R17 [26], and α-synuclein [27], the properties of these IDPs were listed in Appendix A. The detailed simulation conditions for these IDPs and structured proteins are listed in Table 2, which are consistent with corresponding experimental conditions. All simulations are conducted in AMBER18 [28] with ESFF1 force field [29] for protein and TIP3P [30] solvent model, whose initial structures are the same as those in the work of Mu et al. [13]. As the calculated radius of gyration for MD trajectories agreed with experimental values, these test systems were converged and correctly captured the features of disordered proteins [13,15].

For MD simulation preparation, we first packed IDPs in cubic boxes using tleap in AmberTools [28], adding water molecules and neutralizing the system with Na^+^ and Cl^−^ ions. Extra ions were then added to achieve ionic strength in accordance with experimental conditions. Energy minimization was performed with 500 steps of the steep descent method and 500 steps of the conjugate gradient method. A total of 10 picoseconds of canonical ensemble (NVT) MD simulations were conducted to heat the systems. Finally, these systems were switched to NPT and simulated three times independently under the pmemd.cuda acceleration [31]. The electrostatic interactions were calculated using the Particle Mesh Ewald (PME) algorithm [32].

### 4.2. PDB Data Extraction and Preprocessing

Protein trajectories in AMBER trajectory formats were converted into PDB format by CPPTRAJ [33], stripping all solvent molecules and ions. A total of 50,000 frames at 20 ps intervals were saved for each protein. Then, we used the Biobox library in Python to align all frames to the first one according to the RMSD of backbone atoms [34]. The Cartesian coordinates of these heavy atoms were flattened into 1D-vectors (i.e., a coordinate matrix of shape [N,3] is compiled into a vector of length 3N) and further normalized using MinMax scaling as Equation (1):(1)$$c_{i}^{normed}=\frac{c_{i}-min(c)}{max(c)-min(c)}$$
where c refers to the input vector and $c_{i}$ is the i-th coordinate of the vector. Normalization of input data accelerates model convergence and improves the model’s accuracy. As the minimum and maximum of the normalized data are preserved, data can be back-transformed precisely for analysis.

### 4.3. Variational Autoencoder Design

A major purpose of training autoencoders with MD trajectories is that latent variables are effectively solved while encoding. These variables can be used as collective variables (CVs) in meta-dynamic simulations [35], which is feasible for exploring conformation space. AEs are a kind of unsupervised deep learning model whose architecture includes two parts, encoders and decoders [36]. Encoders map the input data (e.g., flattened Cartesian coordinates) to a low-dimensional latent space, while decoders generate high-dimensional data (e.g., reconstructed coordinates) from the encoder-reduced latent vectors with a completely symmetrical structure. However, evidence shows that the latent space in AEs trained on protein conformations fails to capture useful low-dimensional representation [37,38]. Besides, classical AEs are trained with discrete data, which makes it difficult to generate novel conformations of enough credibility. Gupta et al. [12] calculated a multivariate Gaussian distribution in latent space with AEs to sample new IDP conformations. Nevertheless, the Spearman correlation coefficients between the distribution of AE-generated conformations and that of original ones show that the latent distribution of AE is sometimes far from the Gaussian distribution.

To address these problems, we built Variational Autoencoder (VAE) models (Figure 7). Compared with the AE model, VAE adds a variational layer between the encoder and decoder, constraining latent vectors to follow a specified distribution (e.g., Gaussian distribution) during optimization. At first, the encoder of VAE is a deep neuron network (DNN), with a rectified linear unit (ReLU) as an activation function, which bottlenecks the input x to a two-dimension latent space z. Then, the latent space is perturbed by Gaussian noise in variational layer to sample z′, as described by Kingma and Welling [39]. Next, the decoder mirrors the encoder, mapping z′ to the final output x′.

VAE constructs continuous latent space, unlike the discrete one of AE, to sample similar conformations to the input ones. The test results on various protein systems indicate that the performance of VAE is better than AE’s under various criteria, which will be presented in the Result.

### 4.4. VAE Training

In this implementation, we developed the AEs and VAEs in Python3.8 using Keras (https://keras.io/ (accessed on 23 May 2022)) with Tensorflow backend v2.4.1 (https://www.tensorflow.org/ (accessed on 23 May 2022)) [40]. The loss function that assesses the VAE model is calculated by summing two parts: reconstruction loss ($L_{R}$) and Kullback–Leibler divergence loss ($L_{KL}$) with Equation (2):(2)$$L_{VAE}=L_{R}+L_{KL}$$

One of the two parts, reconstruction loss $L_{R}$, is the mean squared error between predicted data x’ and the original input x, which quantifies how well the original data x is approximated by VAE, evaluating the ability of VAE to reconstruct IDP conformations with Equation (3):(3)$$L_{R}=E[||x^{\prime }-x||]=\frac{1}{n}\sum_{i=1}^{n}(x_{i}^{\prime }-x_{i}{)}^{2}$$

The second, Kullback–Leibler divergence loss $L_{KL}$, is based on variational inference on latent space of VAE, which takes the latent space priors that sample z′ into consideration with Equation (4):(4)$$L_{KL}=E\left[\frac{1+log\sigma (z{)}^{2}-\mu (z{)}^{2}-\sigma (z{)}^{2}}{2}\right]$$
where μ and σ, separate latent vectors transformed from an encoder, respectively estimate the mean and standard deviation of the Gaussian prior applied on the latent space. The composite VAE loss function allows the model to sample novel conformations of high credibility with the same priors.

### 4.5. Evaluation Criteria Calculation

Several previous studies have systematically designed the evaluation system for AEs and VAEs on reconstructing the nonlinear representation of tertiary protein structures [41,42,43]. Here we applied a similar strategy to quantify the performance of AE and VAE. There are four evaluation indicators: root-mean-square deviation (RMSD), Spearman correlation coefficient, chemical shift, and radius of gyration (Rg).

### 4.6. Root-Mean-Square Deviation (RMSD)

RMSD quantifies the differences between generated conformations and the input ones. RMSD is calculated as Equation (5):(5)$$RMSD=\sqrt{\frac{1}{N}\sum_{i=1}^{N}{\left(r_{i}^{0}-r_{i}\right)}^{2}}$$
where $r_{i}^{0}$ and $r_{i}$ are Cartesian coordinates of the i-th atom from reference $r^{0}$ and structure r, respectively. N is the number of atoms. Here the coordinates of structure r are all back transformed into Cartesian space, and all heavy atoms in the protein were considered when calculating RMSD.

### 4.7. Spearman Correlation Coefficient

The Spearman correlation coefficient quantitatively reflects the difference in data distribution between generated coordinate vectors and original ones. A higher coefficient indicates that the original space is better preserved by the model. The Spearman correlation coefficient is calculated as Equation (6):(6)$$\rho =1-\frac{6\sum d_{i}^{2}}{n(n^{2}-1)}$$
where $d_{i}$ and n are different in paired ranks and sample size, respectively. The calculation of the Spearman correlation coefficient is conducted with SciPy [44].

### 4.8. Chemical Shift

The secondary chemical shift of Cα can be an index for experimental validation. SPARTA+ [14] is used to calculate the chemical shifts of generated proteins and MD trajectories, and the source of experimental chemical shifts is listed in Appendix A.

### 4.9. Radius of Gyration

The radius of gyration (Rg) is a global structural character that correlates with the length of the protein. Here, MDTraj [45] was used to calculate the Rg of generated proteins and MD trajectories, and the experimental Rg is listed in Appendix A.

### 4.10. Refinement and Visualization of Generated Structures

VAE-generated conformations may exist considerable violations in bond length and angle, so it is necessary to complete and refine the resulting structures. Firstly, tleap was used to add missing hydrogens. Then we subjected the structures to energy minimization in water. This process included 250 steps of the steep descent method and 250 steps of the conjugate gradient method. The force field used was AMBERff14SB [46], and the solvent model was TIP3P.

To obtain representative conformations for visualization, we clustered the refined structures according to their all-against-all RMSD. Here we used the method of k-centroid with the kclust tool provided in MMTSB Tool Set [47] and ensured that the top three clusters covered over 80% of the conformations. The structures selected as centroids of the top three categories are rendered and visualized in Pymol (http://www.pymol.org/pymol (accessed on 1 April 2021)).

## Figures and Tables

**Figure 1 ijms-24-06896-f001:**
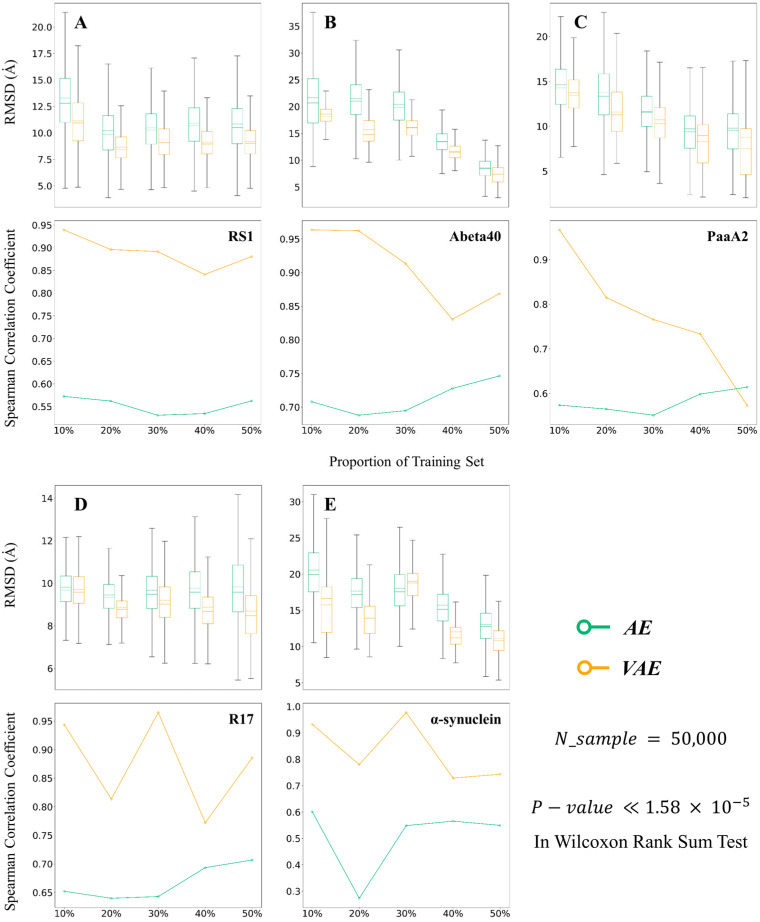
RMSDs and the Spearman correlation coefficients between VAE- or AE-generated IDP conformations and original ones. Dashed lines in boxplots represent mean RMSD. Here, both models are trained on 10% to 50% of conformations separately and are evaluated on the test set. (**A**) RS1. (**B**) Abeta40. (**C**) PaaA2. (**D**) R17. (**E**) α-synuclein.

**Figure 2 ijms-24-06896-f002:**
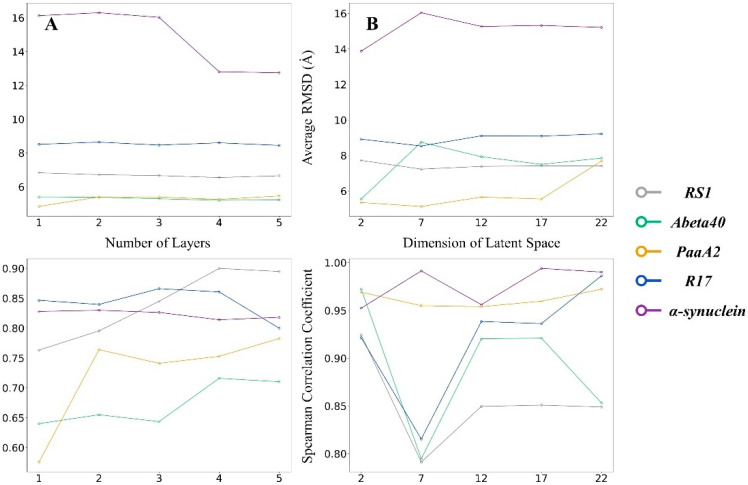
Average RMSD and Spearman correlation coefficient between IDP conformations generated by VAEs of different structures and corresponding MD trajectories. (**A**) RMSDs and Spearman correlation coefficients on VAEs of different layers (1 to 5 layers at 1 interval). (**B**) RMSDs and Spearman correlation coefficients on VAEs whose latent spaces are different sizes. (2 to 22 dimensions at 5 intervals).

**Figure 3 ijms-24-06896-f003:**
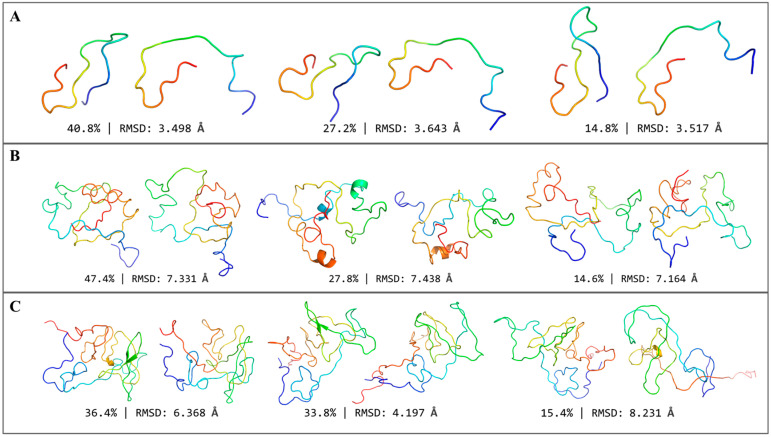
VAE-generated structures (right) that are centroids of the top 3 clusters compared to conformations of the minimum RMSD in MD trajectories (left), blue parts are N-terminus, and red parts are C-terminus here. (**A**) RS1. (**B**) R17. (**C**) α-synuclein.

**Figure 4 ijms-24-06896-f004:**
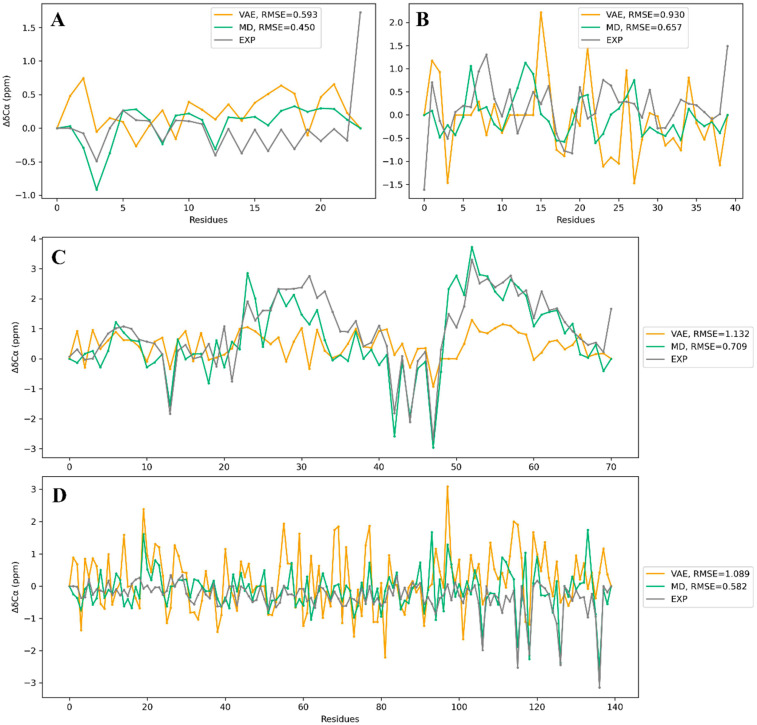
Chemical shifts of VAE-generated conformations and MD trajectories compared to experimental data. Experimental chemical shift data of R17 are missing, so R17 is not displayed here. (**A**) RS1. (**B**) Abeta40. (**C**) PaaA2. (**D**) α-synuclein.

**Figure 5 ijms-24-06896-f005:**
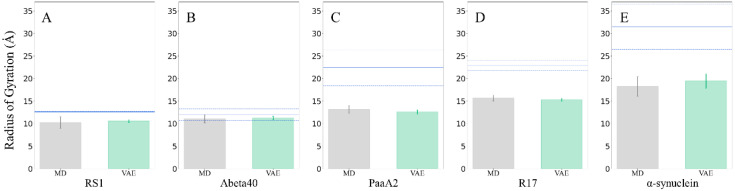
Radius of gyrations for VAE-generated conformations (green) and MD trajectories (grey) compared to experimental data (blue line, dashed lines represent experimental error). (**A**) RS1. (**B**) Abeta40. (**C**) PaaA2. (**D**) R17. (**E**) α-synuclein.

**Figure 6 ijms-24-06896-f006:**
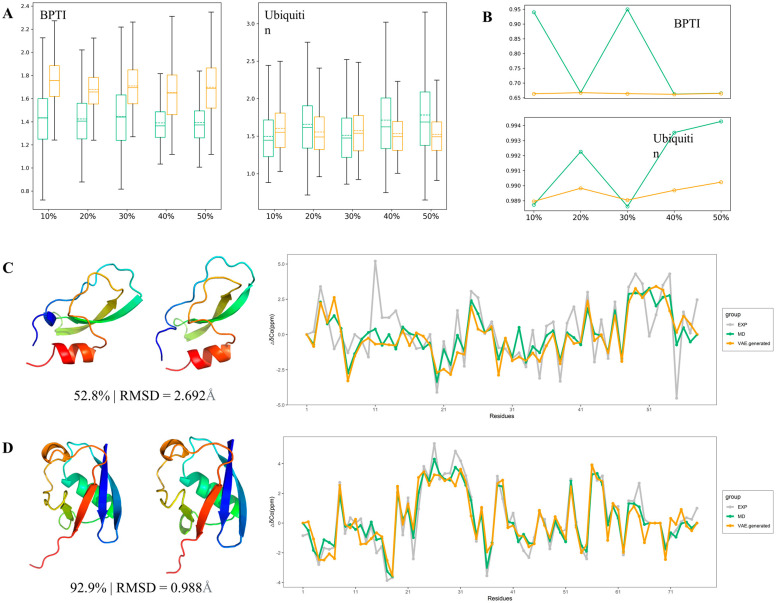
Tests of VAEs’ performance on structured proteins. (**A**) RMSD between original conformations and generated ones. (**B**) Spearman correlation coefficient. (**C**) structures and chemical shifts of BPTI. (**D**) structures and chemical shifts of ubiquitin.

**Figure 7 ijms-24-06896-f007:**
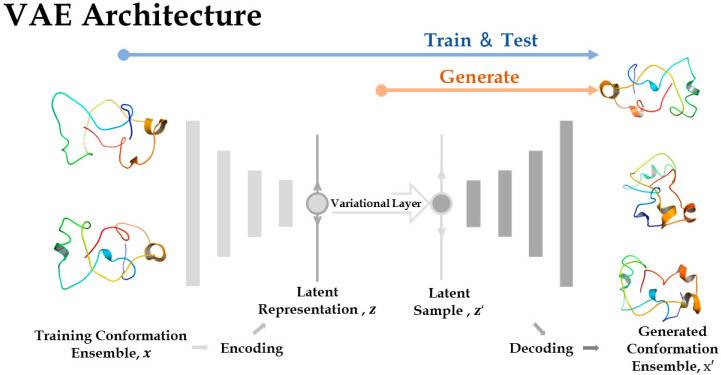
An overview of the architecture of variational autoencoders.

**Table 1 ijms-24-06896-t001:** Structure of AEs and VAEs where all encoders and decoders are deep neuron networks. Here $N_{Res}$ denotes the number of residues of the input protein.

	Encoder	Latent Space	Decoder
AE	1024, 512, 256	$2/3N_{Res}$	256, 512, 1024
VAE	1024, 256, 64, 16	2	16, 64, 256, 1024

**Table 2 ijms-24-06896-t002:** Protein systems involved in this study. Molecular dynamic simulations are conducted according to the temperature and ion strength listed here.

Protein Systems	Length, aa	Temperature, K	Ion Strength, mM	Time (ns)
Intrinsically Disordered Proteins				
RS1	24	298	150	1000
Abeta40	40	277	20	
PaaA2	71	298	500	
R17	100	295	108	
α-synuclein	140	285.5	150	
Ordered Proteins				
Ubiquitin	76	298	50	1000
BPTI	58	300		

## Data Availability

The VAE codes are deposited at https://github.com/Junjie-Zhu/VAE.git (accessed on 4 March 2023).

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
