# Peer review of "Enhancing Conformational Sampling for Intrinsically Disordered and Ordered Proteins by Variational Autoencoder"

_ijms, 2023, doi:10.3390/ijms24086896_

Round 1

Reviewer 1 Report

The authors introduced the variational autoencoder to enhance the sampling of IDPs from molecular dynamics simulations.  The work followed a recent work Zhou and coworkers using a generic autoencoder with the same purpose and showed reasonable improvement to the original method.  I provide my detailed comments below.

Fig. 2, when comparing AE and VAE, the authors used the difference between two values from AE and VAE both with large error bars.  I guess if the authors calculate the Z-score of the two values, they might be very close.  There seems to be a systematic trend that VAE is better than AE, but with these error bars, a statistical test can be used to show if the difference is significant.

The x-axis of Fig. 2 needs to be explained, is that percentage of conformation used for training?  The name of the protein in addition to just ABC can be added as a figure legend.

The performance can somehow depend on how the postprocessing (energy minimization) was done.  The authors used 250 steps of steep descent and 250 steps of conjugate gradient.  They might want to comment on that special combination and how these methods could affect the results.

Fig. 3, it might be better plotting RMSD/correlation coefficient as a function of layer number for each protein.  Currently it's difficult to see there is a turning behavior along the layer number.

Fig. 4, same comments of Fig. 3 applies to Fig. 4.  And Fig.3 and 4 can be merged.

Fig. 6, it's known the errors of calculating chemical shift is large.  The authors might want to add the error bars onto Fig. 5.  Even considering the error bars, VAE seems to be not able to reproduce some of the secondary structures, e.g. C residue 50 to 60.  The authors need to show how AE performs for reproducing the chemical shift/secondary structure.  And for 5.D, please double check the data from VAE.  One more straightforward comparison is to directly compare the secondary structure propensity per residue for ensembles generated by VAE and MD, instead of using chemical shift from experiment.  

The number of significant figures or decimal places reported for a specific variable are quite different in some cases.  For e.g. RMSD 38.625 and 7.53.

Current figures are in low resolution. Fonts in axis label of multiple figures are too small.

This is beyond the scope of this manuscript.  It's question of interest that how much could machine learning methods be used to extend the molecular dynamics trajectories from short trajectories.  Currently the authors used 1 microsecond trajectories.  Some existing trajectories generated by Anton in public domain could be good test sets.

Author Response

  • Figure 2, when comparing AE and VAE, the authors used the difference between two values from AE and VAE both with large error bars. I guess if the authors calculate the Z-score of the two values, they might be very close.  There seems to be a systematic trend that VAE is better than AE, but with these error bars, a statistical test can be used to show if the difference is significant.

Author Reply: Here we mentioned in article that we did make a Wilcoxon rank sum test and get a p-value ≪1.58e-5 (P12, line 2), which mean that the mean RMSDs and spearman coefficients of VAE are significantly lower than those of AE. We updated with the number of samples and maximum p-value observed in Wilcoxon test annotated.

  • The x-axis of Fig. 2 needs to be explained, is that percentage of conformation used for training? The name of the protein in addition to just ABC can be added as a figure legend.

Author Reply: We added the explanation of x-axis in Fig.2, and names of protein are added on the mid-right top of figures separately.

  • The performance can somehow depend on how the postprocessing (energy minimization) was done. The authors used 250 steps of steep descent and 250 steps of conjugate gradient.  They might want to comment on that special combination and how these methods could affect the results.

Author Reply: Our refinement method refers to the research of Gupta et al. (https://doi.org/10.1038/s42003-022-03562-y) While conducting refinement, we referred to the energy minimization in MD simulation, which consisted of 500 steps of steep descent and 500 steps of conjugate gradient. We found that using half of that was enough for refining the generated conformations as this combination corrected the bond and angle violations. This refinement did not affect the overall protein structure while it slightly reduced the RMSD (~0.3A).

  • Figure 3, it might be better plotting RMSD/correlation coefficient as a function of layer number for each protein.  Currently it's difficult to see there is a turning behavior along the layer number. Fig. 4, same comments of Fig. 3 applies to Fig. 4.  And Fig.3 and 4 can be merged.

Author Reply: We have followed the reviewer’s suggestion and fixed the Fig. 3 and Fig. 4 in the original manuscript. Fig.3 in the revised manuscript is the combination of the two..

  • Figure 6, it's known the errors of calculating chemical shift is large. The authors might want to add the error bars onto Fig. 5.  Even considering the error bars, VAE seems to be not able to reproduce some of the secondary structures, e.g. C residue 50 to 60.  The authors need to show how AE performs for reproducing the chemical shift/secondary structure.  And for 5.D, please double check the data from VAE.  One more straightforward comparison is to directly compare the secondary structure propensity per residue for ensembles generated by VAE and MD, instead of using chemical shift from experiment.

Author Reply:  Here we displayed the chemical shift because it’s a common index in evaluating MD and relevant tools, for example Song, D., et al (doi: 10.1021/acs.jcim.0c00059.), Jiang, Y., et al (doi: 10.1039/d2cp04501j.). Though there indeed exist calculation errors, chemical shift is an essential index produced by experiments while we did not get the secondary structure propensity from experiments. What’s more, we can draw a conclusion from the data that VAE trained on MD trajectories did not amplify the error between MD and experimental data much.

AE’s performance on reproducing chemical shifts were displayed in the article of Gupta et al (https://doi.org/10.1038/s42003-022-03562-y), and we use chemical shift only to verify the effect of VAE achieved on experimental results, not mean to compare with AE, so we did not display data from AE in the original manuscript.

Song, D., et al., Environment-Specific Force Field for Intrinsically Disordered and Ordered Proteins. J Chem Inf Model, 2020. 60(4): p. 2257-2267.

Jiang, Y. and H.F. Chen, Performance evaluation of the balanced force field ff03CMAP for intrinsically disordered and ordered proteins. Phys Chem Chem Phys, 2022. 24(48): p. 29870-29881.

  • The number of significant figures or decimal places reported for a specific variable are quite different in some cases. For e.g. RMSD 38.625 and 7.53.

Author Reply: Decimal places have been unified for specific variables in the full text, two decimal places are kept for RMSDs and three for spearman correlation coefficients.

  • Current figures are in low resolution. Fonts in axis label of multiple figures are too small.

Author Reply: In the revised manuscript, we refined the figures.

  • This is beyond the scope of this manuscript. It's question of interest that how much could machine learning methods be used to extend the molecular dynamics trajectories from short trajectories.  Currently the authors used 1 microsecond trajectories.  Some existing trajectories generated by Anton in public domain could be good test sets.

Author Reply: It’s a rather great idea to transfer VAE and relevant methods to much longer trajectories! In this research, we uniformly use 1 microsecond trajectories to ensure comparability between different IDP systems In the following work, we will adopt this suggestion. Thank you for your precious advice!

Reviewer 2 Report

This manuscript describes the use of a variational autoencoder to generate conformational ensembles for intrinsically disordered proteins.

The underlying idea is nice and fits well into the recent trends of biomolecular computing. However, I have some serious concerns about the execution and interpretation of the work.

The most important aspect would be to clearly state the goal of the present implemenation. In my understanding:

1) VAE is trained using the first part of a trajectory, separately for each protein

2) VAE is evaluated on the second part of the MD trajectory - these are the results that are presented

3) VAE can generate novel conformations if perturbed - It is not clear from the present version of the manuscript which results, if any, correspond to these novel conformations

My most serious concerns about the approach and the description is that the conformational space represented by the input and output structures has to be analyzed in deeper detail. The authors mention clustering but do not present its results in  detail. In this respect I am skeptical about the clustering used as it is based on RMSD to the starting structure, which is a distance-type metric and two structures with the same RMSD relative to the staring one might actually be highly different. This should be either clarified or conducted dfferently, e.g. based on principal component analysis. The authors do not even quantify the Rg values they claim to be consistent with experimental values in the literature. These questions are of crucial importance as MD simulations tend to arrive at a compact conformation for IDPs even when starting from an extended initial conformation and this also means that the variation and conformational sampling in a long MD run might be far from even, with the initial part containing more extended states and the latter ones fluctuating around a more compact average structure. This also limits the conformational sampling achievable by conventional MD and might affect the splitting of the trajectories for training and test as described. Thus, without detailed insight into this sampling aspect it is virtually impossible to judge the performance and usability of the constructed VAE.

Additional issues:

- It is not clear - at lest for a structural biologist - what the author mean by "unintended high-dimensional samples" in the paper and why their generation is an issue.

- In Figure 5, centroids of clusters are compared to MD-derived conformations, and it is not clear how the latter ones were selected. 

- In connection with this, the exact name and a structural description of the test systems including secondary structure or preferences, which are especially relevant e.g. for PaaA2 which has strong helical preferences.

- The authors performed geometry optimization of the VAE-derived structures. Were any structure validation steps conducted on the results to show that the optimization resulted in realistic conformations? It is also not clear why heavy atoms might be missing from VAE-generated structures.

- The calculated secondary chemical shifts are not informative in their present form. The reproduction of local preferences and/or some measure on improvement of the correspondence (relative to a well chosen reference maybe) would be required. E.g. in Figure 6C the shifts at the C-terminal of PaaA2 clearly need explanation in terms of structure. For secondary shifts, RMS errors are low because the secondary shift values themselves are low, but the trends do not seem to be reproduced in many cases.

- Table 1: please provide justification for the different parameters used in MD runs Kindly state unambiguously if 1000 ns was the length of simulation in all cases. 

- Please replace "Structural proteins" with "structured" or "globular" proteins.

- In the sentence: "in Pythonto align all frames to the first one according to RMSD of backbone atoms[26]"

- Please add the space after "Python". RMSD is a measure of the goodness of the resulting fit, please be rephrase the sentence and state whether you have used least-squares fitting.

- The use of normalized coordinates is not clear. Were all the heavy atom coordinate vectors complide into a single vector with 3n dimensions, where n is the number of heavy atoms? 
  Normalization in Eq (1) - will you be able to reproduce the original vectors with acceptable numerical precision?
  It is not clear in which subsequent steps the normalized coordinates are ues, e.g. in Eq (3)?
  In Eq. (5), are the normalized coordinates back.transformed into Cartesian space? 
   Eq. (7): are the (normalized) coordinate vectors ranked? If yes, how exactly?

- What was the starting conformation for all the MD runs? How was the structure for IDPs obtained or generated?

- Please use 'first part' instead of 'previous part' in the sentence describing the split of the trajectories to training and test parts.

- Please explain Figure S1. For example, for R17 (panel D) both the RMSD and the Spearman's coefficient are increasing, although - without further explanation - the RMSD is expected be low when the structures are reproduced. In addition, some of the diagrams seem to ave a large stochastic variation instead of monotonous improvement and it is not clear whether it reflects a problem in the setup/execution or expected behavior.

- In Figs 4 and 5, connecting the points is confusing as the data refer to different structures.

- In figure 2 and 7, I guess the box plots show the distribution of RMSD values calculated between each member of the two ensembles compared(?) Yet on Fig. S1, single values (averages?) for RMSD are reported. Please state explicitly what values are shown in each panel/figure. 

- The scales on the Y axis on all diagrams are arbitrary. At least within one figure they should be scaled to allow visual comparison.

Author Response

  • VAE can generate novel conformations if perturbed - It is not clear from the present version of the manuscript which results, if any, correspond to these novel conformations.

Author Reply:  In this study, VAE can be proven to generate novel conformations. The novel conformations here mean these conformations different from the training set. Thus, it could realize the effect of enhanced sampling for the conformation study of IDPs.

We used the trained VAE to generate conformations of target IDPs and calculate the RMSD between these conformations and the original ones. We visualized several conformations and found the most similar conformations in MD trajectories, which are similar to original ones. These cases proved that through training process, VAE actually “learnt” the corresponding properties of  IDPs structure.

Meanwhile, VAE also generated some conformations that are of high RMSD against original ones, we defined this part of conformations as novel ones. While they differed from the input original conformations, they should reserve essential properties of IDPs just as their visualization results do.

  • The authors mention clustering but do not present its results in detail. In this respect I am skeptical about the clustering used as it is based on RMSD to the starting structure, which is a distance-type metric and two structures with the same RMSD relative to the staring one might actually be highly different.

Author Reply:  We used clustering to make the visualized conformations more representative, and to ensure that they were distinct from each other.

Firstly, what we have to admit is that the question you mentioned is almost inevitable in clustering. Even if we use different clustering methods like using phi-psi angles distribution, we would meet the same trouble that some different conformations are clustered together.

However, what we can make sure is that the conformations from different clusters were differ from each other. We could get more diverse and representative conformations by clustering, instead of randomly selecting. With these diverse conformations, we could claim that VAE generated different patterns of IDPs, and the model is useful.

  • The authors do not even quantify the Rg values they claim to be consistent with experimental values in the literature.

Author Reply: We calculated Radius of gyrations of VAE-generated conformations and MD trajectories compared to experimental data. The results is shown in Figure 6  in the revised manuscript.

  • MD simulations tend to arrive at a compact conformation for IDPs even when starting from an extended initial conformation and this also means that the variation and conformational sampling in a long MD run might be far from even, with the initial part containing more extended states and the latter ones fluctuating around a more compact average structure.

Author Reply:  The model was trained on the first part of MD trajectories (e.g. the first 30% of the trajectories is equal to samples from a 300ns MD simulation), in this part the conformations are more extended, as you have mentioned. That's exactly what we want that VAE was trained on these extended IDP conformations with more characters.

We used the latter part of conformations, which are more compact average structures for test. The results of test show that some of generated conformations are similar to ones from the latter part in MD (as visualization shown), while there also existed conformations that are highly different from original input (as RMSD indicated). The former proved that the VAE-generated conformations are credible, and the latter proved that there should be novel conformations that are extended as those in the training set.

  • It is not clear - at lest for a structural biologist - what the author mean by "unintended high-dimensional samples" in the paper and why their generation is an issue.

Author Reply: “unintended” here means novel samples, corresponding formulations are revised as novel high-dimensional samples”.

  • In Figure 5, centroids of clusters are compared to MD-derived conformations, and it is not clear how the latter ones were selected.

Author Reply: We calculated the minimum RMSD to get the most similar conformations in MD trajectories. We filled in this detail in the revision manuscript.

  • In connection with this, the exact name and a structural description of the test systems including secondary structure or preferences, which are especially relevant e.g. for PaaA2 which has strong helical preferences.

Author Reply:  Considering the input of our model is only Cartesian coordinates, we didn’t include these details in our article. Detailed information about these proteins is described in references 13-17:

  • The authors performed geometry optimization of the VAE-derived structures. Were any structure validation steps conducted on the results to show that the optimization resulted in realistic conformations? It is also not clear why heavy atoms might be missing from VAE-generated structures.

Author Reply: We think we have provided enough evidence to prove the ability of VAE, such as Rg, chemical shifts and visualization of the structures. Maybe there were some sentences in the original manuscript that the reviewers misunderstood. VAE-generated structures do not lose any heavy atoms, and we’ve revised relative sentences.

  • The calculated secondary chemical shifts are not informative in their present form. The reproduction of local preferences and/or some measure on improvement of the correspondence (relative to a well chosen reference maybe) would be required. E.g. in Figure 6C the shifts at the C-terminal of PaaA2 clearly need explanation in terms of structure. For secondary shifts, RMS errors are low because the secondary shift values themselves are low, but the trends do not seem to be reproduced in many cases.

Author Reply: Secondary chemical shift is used mainly to measure the difference between calculated conformation and true conformation in experiment. We can find that MD itself owns error against experimental data. Here we exploit chemical shift to confirm that our model trained on MD trajectories did not amplify such error much. In some case, chemical shift calculated indeed deviates from experimental data, that may result from the novel extended conformations generated by VAE. For example, PaaA2 has strong helical preferences in its stable state, while VAE can generate conformations that are in transition states and own less helices. To describe the generated conformations more detailly, we added Rg as another index besides chemical shifts.

  • Table 1: please provide justification for the different parameters used in MD runs Kindly state unambiguously if 1000 ns was the length of simulation in all cases.

Author Reply: Relevant explanations has been added.

  • Please replace "Structural proteins" with "structured" or "globular" proteins.

Author Reply: We have replaced "Structural proteins" with “structured proteins” as advised.

  • In the sentence: "in Pythonto align all frames to the first one according to RMSD of backbone atoms[26]" Please add the space after "Python". RMSD is a measure of the goodness of the resulting fit, please be rephrase the sentence and state whether you have used least-squares fitting.

Author Reply: RMSD in our work is to evaluate if two conformations are similar, so least-squares fitting is not required here. We have corrected some clerical errors in the original manuscript as suggested above.

  • The use of normalized coordinates is not clear. Were all the heavy atom coordinate vectors complide into a single vector with 3n dimensions, where n is the number of heavy atoms?

Normalization in Eq (1) - will you be able to reproduce the original vectors with acceptable numerical precision?

It is not clear in which subsequent steps the normalized coordinates are ues, e.g. in Eq (3)?

In Eq. (5), are the normalized coordinates back.transformed into Cartesian space?

Eq. (7): are the (normalized) coordinate vectors ranked? If yes, how exactly?

Author Reply: We added the Description about the use of normalization and how heavy atoms are processed.

Normalization is conducted on the 3n vectors, right before putting into model. Normalization do not affect precision for the scaler is recorded and used for back transformation.

Coordinates are back transformed.

Wilcoxon test is conducted with SciPy, a package of Python. Ranks is conducted on MinMax scaled input data and the direct outputs of model. Actually, back-transformed makes no effect on rank sum test.

  • What was the starting conformation for all the MD runs? How was the structure for IDPs obtained or generated?

Author Reply: Starting conformations are the same to Mu, J. et al. (doi: 10.1021/acs.jcim.1c00407.). We used the same structures systems.

Mu, J., Z. Pan, and H.F. Chen, Balanced Solvent Model for Intrinsically Disordered and Ordered Proteins. J Chem Inf Model, 2021. 61(10): p. 5141-5151.

  • Please use 'first part' instead of 'previous part' in the sentence describing the split of the trajectories to training and test parts.

Author Reply: We have revised as advised.

  • Please explain Figure S1. For example, for R17 (panel D) both the RMSD and the Spearman's coefficient are increasing, although - without further explanation - the RMSD is expected be low when the structures are reproduced. In addition, some of the diagrams seem to ave a large stochastic variation instead of monotonous improvement and it is not clear whether it reflects a problem in the setup/execution or expected behavior.

Author Reply: The training of VAE includes an adversarial process of fitting gaussian distribution in latent space and reconstruct to the exact target. Here the RMSD and spearman correlation coefficients in Figure S1 were not recorded in one training process, but were recorded from models trained for certain epochs. The RMSDs were between each members generated conformations and original ones. We took the average RMSD here to see how much generated conformations deviated from original ones, and to see if it tended to stabilize. From the result, we found that spearman coefficients tended to stabilize at a rather high value, meaning that the reproduced distribution was similar to the original, while RMSD did not stabilize well. It may result from the randomness of sampling from gaussian distribution. For the variation, it’s partly caused by randomness between each training process. Beyond such variation, the result actually reflected that VAE generate diverse and credible conformations.

  • In Figs 4 and 5, connecting the points is confusing as the data refer to different structures.

Author Reply: We have refined Figs 4 and 5.

  • In figure 2 and 7, I guess the box plots show the distribution of RMSD values calculated between each member of the two ensembles compared(?) Yet on Fig. S1, single values (averages?) for RMSD are reported. Please state explicitly what values are shown in each panel/figure.

Author Reply: The distribution is calculated between each member of the two ensembles compared. The single values are averages exactly; figure S1 have been revised.

  • The scales on the Y axis on all diagrams are arbitrary. At least within one figure they should be scaled to allow visual comparison.

Reply: We have revised the scales of Y axis in the revised manuscript. Thank you for your advice.

Reviewer 3 Report

The manuscript submitted by Hai-Feng Chen and colleagues describes an artificial intelligence application to the problem of sampling protein conformational space.

The Authors describe a series of classical MD simulations of 7 proteins, five conformationally disordered and two globular. The structural snapshots are the input of a modified autoencoder, the variational autoencoders, which is able to reproduce the conformational diversity of these proteins better than simple autoencoders.

The performance of these machine learning techniques is evaluated by means of the root-mean-square-distance, the non-parametric Spearman correlation coefficient, and the computed chemical shifts.

The manuscript is well written and quite accessible also to “molecular” scientists. Some modifications are nevertheless advisable.

Important: the usefulness of this approach is insufficiently documented. What is the reason to “simulate” a MD experiment, if everything is based on the data produced by MD?

Important: it is necessary to indicate how faster is the generation of protein models with VAE than with classical MD.

Equation (5). Which atoms are considered? Only the Calpha or all the atoms?

Equation (6). It is unclear what quantity is sorted. If it is the distance between equivalent atoms, like in equation (5), the same notation should be used.

Page 5, line 5 from the bottom. The reference Gupta et al. is missing. This occurs also on page 7, line 3 from the bottom.

Figure 2 – and also other figures. It is necessary to indicate the quantity that is plotted along the axis (with the unit measure, if necessary). Axes labels should be readable, now they are too small. In Figure 2 it is also unclear what the horizontal axis means.

Paragraph “Visualization of VAE-generated conformations” line 4. “were” should be “are”.

Paragraph “Visualization of VAE-generated conformations” line 6. The expression “structural region” is not used in molecular biology. “Structured region” is better. Note that this problem exists also in several other parts of the text.

Figure 5. Only three proteins are considered. It would be nice to include also the other two.

Figure 5. Structures of MD and of VAE clusters are shown. Which is left, which is right?

Figure 5A. The three clusters are extremely similar. This is surprising and must be commented.

Figure 5. I suppose that the color code (blu-> red) is the common N- to C-terminus. However, this should be clearly indicated in the caption.

Figure 6. Why R17 is missing?

Figure 6. Protein Abeta40 is in other parts of the manuscript named Ab40 (Table 1, for example). The Authors should be consistent: either they use “beta” or they use the Greek symbol.

Some of the references (ex. 8 and 28) are incomplete.

Author Response

  • The usefulness of this approach is insufficiently documented. What is the reason to “simulate” a MD experiment, if everything is based on the data produced by MD?

Author Reply: Actually our model was trained only with the first part of an MD trajectory, e.g., 30% of a trajectory equaled to a 300ns simulation. From this 300ns simulation we generated conformations that were similar to those in 1 microsecond simulation with low computational cost. This model is of potential in substituting part of MD, thus enhancing the sample efficiency of protein structure study based on MD.

  • It is necessary to indicate how faster is the generation of protein models with VAE than with classical MD.

Author Reply: Tests on generation velocity are now added to the manuscript.

  • Equation (5). Which atoms are considered? Only the Calpha or all the atoms?

Author Reply: All heavy atoms, and relevant statements have been added.

  • Equation (6). It is unclear what quantity is sorted. If it is the distance between equivalent atoms, like in equation (5), the same notation should be used.

Author Reply: Spearman correlation coefficient is a metrics from the Tian, H., et al. (doi: 10.3389/fmolb.2021.781635.). Here the input data and direct output data are used to calculate their ranks.

Tian, H., et al., Explore Protein Conformational Space With Variational Autoencoder. Front Mol Biosci, 2021. 8: p. 781635.

  • Page 5, line 5 from the bottom. The reference Gupta et al. is missing. This occurs also on page 7, line 3 from the bottom.

Author Reply: The missing references have been added.

  • Figure 2 – and also other figures. It is necessary to indicate the quantity that is plotted along the axis (with the unit measure, if necessary). Axes labels should be readable, now they are too small. In Figure 2 it is also unclear what the horizontal axis means.

Author Reply: Fig. 2 has been refined.

  • Paragraph “Visualization of VAE-generated conformations” line 4. “were” should be “are”.

Paragraph “Visualization of VAE-generated conformations” line 6. The expression “structural region” is not used in molecular biology. “Structured region” is better. Note that this problem exists also in several other parts of the text.

Author Reply: All these mistakes have been corrected.

  • Figure 5. Only three proteins are considered. It would be nice to include also the other two.

Author Reply: The other two have been put in supplementary materials (Figure S2).

  • Figure 5. Structures of MD and of VAE clusters are shown. Which is left, which is right?

Author Reply: Right is VAE-generated structures, and left is MD structures. The illustration has been added to the manuscript.

  • Figure 5A. The three clusters are extremely similar. This is surprising and must be commented.

Author Reply: Although they look very similar, that's because RS1 is a rather short IDP, and in fact, they are different both in their N- and C-terminus, and fold to a different angle overall. Also, VAE-generated conformations tend to be unfolded. These clusters represent different unfolded forms of RS1.

  • Figure 5. I suppose that the color code (blu-> red) is the common N- to C-terminus. However, this should be clearly indicated in the caption.

Author Reply: We have indicated this information in the caption.

  • Figure 6. Why R17 is missing?

Author Reply: Because R17 is lack of open experimental data, we did not present its chemical shift here. This has also been explained in the manuscript.

  • Figure 6. Protein Abeta40 is in other parts of the manuscript named Ab40 (Table 1, for example). The Authors should be consistent: either they use “beta” or they use the Greek symbol.
  • Some of the references (ex. 8 and 28) are incomplete.

Author Reply: Modified.

Round 2

Reviewer 1 Report

The authors have addressed all my questions in details.  I recommend the manuscript to publish in IJMS.

Author Response

We polish the language in this version.

Reviewer 2 Report

I would like to thank the authors for their efforts put into the revision. I also would like to make my position clear: I think that the developed method has its merits and could be useful, but I also think that the current level of presentation (which did not improve in important aspects relative to the first version) prevents the paper from having the impact it could have among structural biologists. 

In general, I still miss a detailed and biologically meaningful comparison of the conformational spaces represented by the training, test and generated conformer sets. Without a detailed overview of the main structural features it is not possible to evaluate the performance and thus the usability of the method as it is (even if it has great potential presently or can have in the future).

In their answers the authors claim that when a generated conformer is similar to the test set it proves that the system is trained well and when it is dissimilar it proves that the system is capable of generating diversity. This statement is simply too general to have any scientific merit unless demonstrated in much more detail.

Therefore, I still ask the authors to provide a comparative analysis of some global and local structural features and I still suggest the use of PCA or another method that is more informative than RMSD from the first structure (e.g. all-against-all RMSD could be much better).

Specifically:

- My first point about the demonstration of the different conformations is not satisfactorily addressed, many of my concerns are described above. Visualization of some selected conformers is of course necessary but not sufficient.

- I appreciate the answer to my comment on clustering but again, without any specific demonstration of the clusters the present description is insufficient. Again, a PCA or an all-against-all RMSD could be more informative for comparing the clusters and their members. I would also argue that phi-psi based clustering would also be much more appropriate than the one based on the RMSD from the first structure.

- Showing the Rg values is a good step towards demonstrating the capabilities of the method. However, the scales on the Y axes are again quite different and the difference between experiment and MD/VAE for both PaaA2 alpha-synuclein seems huge and definitely needs to be explained.

- My concern about the division on the MD trajectories to training and test data are confirmed by the authors in their answer. However, no reference to the extended/compact nature of conformations is inserted in the paper, let alone a discussion or (desirably) tests with different training/test setups.

- The authors use now "novel high-dimensional samples" but what exactly "high-dimensional" means is still not entirely clear. I guess they refer to the normalized 3N vectors as "high-dimensional"? This is still not accessible for structural biologists, the major target group of the method.

- The author claim that because the input of their method is coordinates, they can omit structural description of the proteins. However, developing an application for structural biology can not be described and/or interpreted without detailing the structural biology of the systems used for its presentation.

- The authors still do not demonstrate the structural feasibility of the structures generated, although this could be an issue for they approach. They have deleted the reference for missing heavy atoms from the description of using tleap in the refinement stage. Still, a demonstration of some kind of structure validation could be  desirable, if possible (at least to demonstrate that the VAE-generated conformations and not worse in this sense than the MD-derived ones).

- Chemical shifts: I can not accept the authors' reasoning without further analysis. It might well be that they VAE-generated ensemble contains a sub-ensemble that matches the MD-derived and/or experimental shifts well but this can not be judged on the basis of the present plot. Again, the explanation that matches prove learning and mismatches demonstrate novel structures is unacceptable as the presented plots can well be the result of a much less diverse VAE ensemble than the MD-derived one. Secondary chemical shifts are highly sensitive for local structure and if the authors use them for demonstration purpose they should provide a more detailed analysis to show the properties of the VAE-derived structures in this respect. In other words, if VAE-generated structures can not be used to get a better ensemble in terms of correspondence to Rg and chemical shifts than MD is capable of, then the whole point of VAE in its present form is questionable for the structural biology community (even if the development itself can be valid in the field of deep learning).

- I do not understand the authors' reply stating that least-squares-fit is not required. If the RMSD values given are not calculated from fitted structures, they are meaningless. If structure fitting was done, it was definitely using some algorithm which is usually least-squares-fitting. I guess we are talking about different things here which should be clarified.

- Thank you for the clarification on coordinate transformation.

- Thanks for clarifying the starting structures. Because it is of crucial importance, I think it would be beneficial to insert the details into the present manuscript also.

- I accept all the other technical answers, except that the Y axes are still not scaled in many panels. Also, the description on the stabilization on RMSD and correlation could well go into the manuscript.

Author Response

  • My first point about the demonstration of the different conformations is not satisfactorily addressed, many of my concerns are described above. Visualization of some selected conformers is of course necessary but not sufficient.

Author Reply: At first, we do cluster analysis for these conformations, then representative conformations were chosen and displayed.

  • I appreciate the answer to my comment on clustering but again, without any specific demonstration of the clusters the present description is insufficient. Again, a PCA or an all-against-all RMSD could be more informative for comparing the clusters and their members. I would also argue that phi-psi based clustering would also be much more appropriate than the one based on the RMSD from the first structure.

Author Reply: Actually, the tool we used for clustering is an all-against-all RMSD method. (J. Mol. Graph. Model. 2004, 22:377-395). We also add PCA analysis, the results are shown in Figure S3. This figure indicates that VAE could generate diverse conformations. The method of k-centroid with kclust tool in MMTSB Tool Set was used for cluster.

  • Showing the Rg values is a good step towards demonstrating the capabilities of the method. However, the scales on the Y axes are again quite different and the difference between experiment and MD/VAE for both PaaA2 alpha-synuclein seems huge and definitely needs to be explained.

Author Reply: The Y-axes are uniform and revised. We added explanation why MD/VAE deviate from experimental data in Rg. Considering that our VAE model is trained completely based on the structures from the MD simulations, correcting the error between MD and experimental data is the fundamental strategy to further improve the performance of VAE model. Our VAE learns from MD trajectories, and reproducing of MD conformations and sampling similar ones is what it’s supposed to do. Though we can’t reduce the gap between experimental data and our computational data, we do accelerate the process of sampling. Our work indicates that VAE trained with a short MD trajectory reproduces a relatively long trajectory well, while consuming much less time (Table S4). If we want to reduce the gap between MD and experimental data, we can use carefully-selected conformations from MD that’s consistent with the experimental properties in training VAE, but in that form it will be hard for us to evaluate how VAE reproduces MD trajectories, and we can’t get to the conclusion that VAE is of potential in improving the speed MD or even substituting for it.

  • My concern about the division on the MD trajectories to training and test data are confirmed by the authors in their answer. However, no reference to the extended/compact nature of conformations is inserted in the paper, let alone a discussion or (desirably) tests with different training/test setups.

Author Reply: Corresponding explanations have been added to our article. The strategy of using the first part of MD in training VAE is feasible. The deep learning model is just like a child, if you only teach him math, it would be impossible for him to understand chemistry. For VAE, we gave it for training the first part of MD trajectories, which showed larger fluctuation and owned more extended conformations, then it’s possible for VAE to learn from these diverse data, and to generate feasible conformations that are feasible but different from the original ones. If we gave it compact ones which are of low diversity, VAE would be unable to infer how extended conformations look like, thus causing failure in training.

  • The authors use now "novel high-dimensional samples" but what exactly "high-dimensional" means is still not entirely clear. I guess they refer to the normalized 3N vectors as "high-dimensional"? This is still not accessible for structural biologists, the major target group of the method.

Author Reply: We have revised the sentences as you have advised.

  • The author claim that because the input of their method is coordinates, they can omit structural description of the proteins. However, developing an application for structural biology can not be described and/or interpreted without detailing the structural biology of the systems used for its presentation.

Author Reply: Actually, a series of computational method exploited only Cartesian coordinates from the classical MSM, PCA to the recent deep learning methods. Meanwhile, the previous studies did not give structural description of their target proteins. (J. Chem. Theory Comput. 2015, 11: 5513-5524; Eur. Phys. J. B. 2021, 94: 211) Actually, IDPs are few structural descriptions in experiments for the high difficulty of analyzing their structures, as we’ve explained in introduction part in article.

The main target of our work is to provide a much faster tool compared to MD that’s of similar accuracy, and we choose IDP systems of different size, and of course, of different structural properties in our tests. The model performs stable in all these systems, which means that VAE trained with coordinates is not affected by certain structure features (like α-helix). We added a series of structural descriptions in Table S5.

  • The authors still do not demonstrate the structural feasibility of the structures generated, although this could be an issue for they approach. They have deleted the reference for missing heavy atoms from the description of using tleap in the refinement stage. Still, a demonstration of some kind of structure validation could be desirable, if possible (at least to demonstrate that the VAE-generated conformations and not worse in this sense than the MD-derived ones).

Author Reply: Besides visualization, Rg and chemical shift of generated conformations that are not deviated from MD trajectories are both demonstration of the conformations. Relatively low RMSD and high correlation also represent that VAE performs well in conformation generation task. We believe that these indexes are enough for proving the feasibility of generated conformations as they are the same as, or even more than those used in previous works. These indexes should have proved that VAE-generated conformations are identical with the MD-derived ones, while the sampling process only takes several minutes.

For the “missing heavy atoms”, it’s really a mistake in writing article. From the structure of VAE it’s clear that the shape of output vectors is exact the same as the input ones, so there should not be something missing during the generation. The model is trained with all heavy atoms, so the generated conformations transformed from the output vectors definitely contain all these heavy atoms.

  • Chemical shifts: I can not accept the authors' reasoning without further analysis. It might well be that they VAE-generated ensemble contains a sub-ensemble that matches the MD-derived and/or experimental shifts well but this can not be judged on the basis of the present plot. Again, the explanation that matches prove learning and mismatches demonstrate novel structures is unacceptable as the presented plots can well be the result of a much less diverse VAE ensemble than the MD-derived one. Secondary chemical shifts are highly sensitive for local structure and if the authors use them for demonstration purpose they should provide a more detailed analysis to show the properties of the VAE-derived structures in this respect. In other words, if VAE-generated structures can not be used to get a better ensemble in terms of correspondence to Rg and chemical shifts than MD is capable of, then the whole point of VAE in its present form is questionable for the structural biology community (even if the development itself can be valid in the field of deep learning).

Author Reply:  A Gaussian distribution is used in the generation process of VAE, which means that the diversity of conformation can be improved by sampling more conformations or adjusting the parameters in Gaussian distribution. In the article we sampled 5W conformations, the same in quantity with MD trajectories. Our results prove that VAE generates conformations that are similar to those from MD. In Rg and chemical shift VAE does not deviate from MD, and the representative conformations are visualized, proving that they preserve the properties of target IDPs. We also added PCA in Figure S3, which might help prove the diversity of generated conformations.

At the same time, Figure 5 indicates that secondary chemical shifts of VAE model could partly reflect the local structure. For example, PaaA2 has some alpha-helix structure (Figure S2). (Commun. Biol. 2022, 5: 610)

  • I do not understand the authors' reply stating that least-squares-fit is not required. If the RMSD values given are not calculated from fitted structures, they are meaningless. If structure fitting was done, it was definitely using some algorithm which is usually least-squares-fitting. I guess we are talking about different things here which should be clarified.

Author Reply: The conformations are pre-aligned before training or calculating RMSD, which means that the translation and rotation in conformations are removed. Maybe least-squares-fit you mentioned here is the same as alignment in our article. We aligned these conformations with a quaternion-based method but not least-squares-fit, so here we may have talked about different things.

  • Thanks for clarifying the starting structures. Because it is of crucial importance, I think it would be beneficial to insert the details into the present manuscript also.

Author Reply: Modified.

  • I accept all the other technical answers, except that the Y axes are still not scaled in many panels. Also, the description on the stabilization on RMSD and correlation could well go into the manuscript.

Author Reply: Y axes of Rg have been revised. However, the results within the IDP systems but not between the systems are focused on, and these figures are often not in the same line, so we consider it better to remain the present version for RMSD and correlation.

We also revise the article in some details and try our best to fix the language problems. Thanks for the comments.

Round 3

Author Response

  • However, the results of this clustering are practically still not presented. The only quantitative information provided is that the top 3 clusters cover 80% of the conformers. We know nothing about the diversity and size of these clusters. We also do not know whether the VAE-generated structures most similar to the centroids would belong to the same cluster or their RMSD from the centroid falls within the variance characteristic of the cluster. Alternatively, the number of VAE-generated structures close to those in each cluster could be given. As the presented RMSDs (based on the boxplots) are generally large, at least some of these measures would be more informative and could clearly demonstrate that the MD-like conformations are sampled and to what extent. In addition, the PCA shown in Fig. S3 is uninformative – as PCA will highlight the components along which the largest diversity is achieved, the mere fact that the conformations are scattered along these coordinates does not sufficiently demonstrate the diversity of the ensembles.

Reply: K-centroid is used here on 500 generated conformations considering that it’s a rather time-consuming process, and we displayed the proportion of each cluster in figure 4, so that their sizes are clear in the article. PCA has been revised as suggested, and the present form of PCA indicates that VAE generates conformations that are of similar diversity to MD.

  • Were any structure validation steps conducted on the results to show that the optimization resulted in realistic conformations?

Reply: We added validations on phi-psi distribution, which could prove that we generated realistic conformations.

  • The authors’ reply here does not answer my concerns about chemical shifts. Where there is a deviation from the MD-derived shifts as are in the C-terminal region of PaaA2, an explanation would be desirable. The fact that the secondary chemical shifts are closer to zero might indeed result from a more diverse ensemble (that includes conformations that match the MD-derived ones but also many different ones so the shifts average out) but also might come from a less diverse ensemble (that does not contain structures with a C-terminal region similar to those in the MD set, rather displays much less marked structural preferences). This is an important distinction, and without explicit demonstration it is impossible to argue in favor of either of the scenarios with confidence. On the PCA I have made my comments above – without a comparative aspect, the analysis also does not prove the authors’ point.

Reply: We revised the PCA, so that diversity could be proved by PCA and clustering.